

# Development of a surface roughness curve to estimate timing of earthflows and habitat development in the Teanaway River, central Washington State, USA

Sarah A. Schanz[1], A. Peyton Colee[1]

[1]Geology Department, Colorado College, Colorado Springs, 80903, USA

*Correspondence to*: Sarah A. Schanz (sschanz@coloradocollege.edu)

**Abstract.** Salmon habitat is enhanced by the wide valleys and channel heterogeneity created by landslides. Earthflows, which are slow moving and fine-grained mass movements, can further potentially alter habitat by constricting valleys and sustaining delivery of debris and fine sediment. Here, we examine the influence of earthflows on salmon habitat in the Teanaway River basin, central Cascade Range, Washington. We mapped earthflows based on morphologic characteristics and relatively dated earthflow activity using a flow directional surface roughness metric called MADstd. The relative MADstd ages are supported by six radiocarbon ages, three lake sedimentation ages, and 20 cross-cutting relationships, indicating that MADstd is a useful tool to identify and relatively date earthflows, especially in heavily vegetated regions. Our age and MADstd distributions reflect a period of earthflow activity in the mid-Holocene and some sustained movement through the late Holocene that is primed by regolith production in the Pleistocene and early Holocene and triggered by a warm and wet climate during the mid-Holocene. The timing of earthflows is coincident with stabilization of salmon habitat and abundant salmon populations, indicating the fine sediment from earthflows did not negatively impact habitat. Wide valleys formed upstream of valley-constricting earthflows have added habitat zones, which may be of increased importance as climate change causes lower flows and higher temperatures in the Teanaway basin over the next century.

## 1 Introduction

Landsliding transports debris from hillslopes to valley bottoms and can be crucial in creating and maintaining landscape heterogeneity, riparian refuge habitat, and spawning gravels for salmon (Beeson et al., 2018; May et al., 2013). Large wood (LW) transported by mass wasting into the channel results in channel roughness and the formation of resting pools and habitat complexity (Burnett et al., 2007). Deep-seated landslides are associated with wider valleys, a key landscape component for salmon and trout habitat (Beeson et al., 2018; Burnett et al., 2007; May et al., 2013). However, fine debris by landslides can present a habitat challenge as silt clogs the pores between stream cobbles and limits oxygen flow to redds (NFTWA, 1996), and landslides in narrow tributaries may dam the stream and temporarily kill off a small population (Waples et al., 2008). In landslide-dominated landscapes, understanding the history of landsliding is crucial to reconstructing the development and maintenance of salmon habitat.




Previous work on landslides and habitat often focuses on valley width associations, following Burnett et al. (2007)'s identification of valley width and slope as key landscape characteristics controlling salmon habitat. Wide valleys are associated with sinuous and lower slope channels that form pool and riffle sequences (Montgomery and Buffington, 1997) and key summer habitat (Beeson et al., 2018). Wide floodplains host flood refuge habitat and buffer the main channel from episodic

sediment inputs (May et al., 2013). Upstream aggradation from large landslides causes valley widening and results in more productive habitat areas (Korup et al., 2006; May et al., 2013). The landslide-caused wider valleys and associated habitat zones tend to be more connected, especially for winter habitat, than landslide-free landscapes (Beeson et al., 2018) and are often located in high slope reaches that are inaccessible for development and may provide an additional buffer from anthropogenic habitat degradation.


While deep seated landslides can alter the upstream landscape and provide habitat zones for $10^3$-$10^5$ years (Beeson et al., 2018; May et al., 2013), habitat quality downstream degrades for $10^0$-$10^1$ years from the associated sediment and debris delivery (Johnson, 1990). Sediment from landslides can diminish salmon habitat by clogging pore spaces and blocking oxygen flow to redds or by creating a coarse, immobile sediment cover that armors the bed and inhibits redd formation. However, these effects

tend to last for years to decades (Johnson, 1990). Alternatively, debris such as large wood introduced through landsliding provides channel roughness elements needed to develop pools and other habitat refugia (Abbe and Montgomery, 1996; Bigelow et al., 2007; Burnett et al., 2007). While debris and silt may be a habitat disturbance, landslides in small tributaries occur only once in a few salmon generations and recovery tends to be quick (Beechie, 2001).

Although prior work has focused on deep seated landslides and shallow debris flows (e.g., Beeson et al., 2018; Bigelow et al., 2007; Burnett et al., 2007; May et al., 2013), little research has investigated the effect of earthflows on habitat. Earthflows are fine-grained soil mass movements that move meters or less per year and persist for decades to centuries (Hungr et al., 2014). They tend to occur in clay-bearing rocks or weathered volcanic rocks with translational movement, and are commonly reactivated in response to increased precipitation or other disturbances that decrease shear resistance (Baum et al., 2003).

Earthflow movement is correlated to climate and regolith production; over long timescales ($10^1$-$10^4$ years), earthflow movement is limited by the pace of regolith production as transport typically outpaces weathering rates (Mackey and Roering, 2011). At the annual to decadal scale, precipitation variability is correlated with earthflow speed, in which earthflows are observed to speed up—following a lag of several weeks—after seasonal and annual precipitation increases (Coe, 2012; Handwerger et al., 2013). Droughts may prime earthflows by creating deep desiccation cracks that act as water conduits during

ensuing wet conditions (McSaveney and Griffiths, 1987). Similar to deep seated landslides, earthflows can cause upstream channel aggradation and valley widening; Nereson and Finnegan (2018) note an order of magnitude increase in valley width upstream of the Oak Ridge earthflow.



Due to their persistence, earthflows can be major sources of sediment to channels, and therefore a significant disturbance to
salmon habitat. Earthflows in the Eel River basin, although covering only 6% of the basin, account for half of the regional
denudation rate with approximately 19,000 t/km/yr of sediment produced (Mackey and Roering, 2011). In stream sediment
production is unsteady as annual to decadal precipitation conditions and sediment supply cause intermittent movement—and
habitat disturbance—over the decades to centuries that the earthflow is active (Guerriero et al., 2017; Mackey and Roering,
2011). Additionally, earthflows can temporarily transition to debris flows, resulting in rapid transport of weathered material
and debris to the channel (Malet et al., 2005).

Here, we examine the timing of Holocene earthflows in the Teanaway Basin of the central Cascade Range of Washington
State, located in the northwest corner of the continental USA, where Pleistocene glaciations once controlled salmon habitat
but now episodic events such as landslides, fires, and floods dominate the disturbance regime (Waples et al., 2008). Geologic
mapping of the region and recent lidar reveals extensive landsliding in the form of earthflows (Quantum Spatial, 2018, 2015;
Tabor et al., 1982), but the cause and timing of these earthflows is unknown. We develop a relative dating curve for earthflows
and apply it to the Teanaway basin to determine when the earthflows were active and discuss how this affected valley width
and salmon habitat during the Holocene.

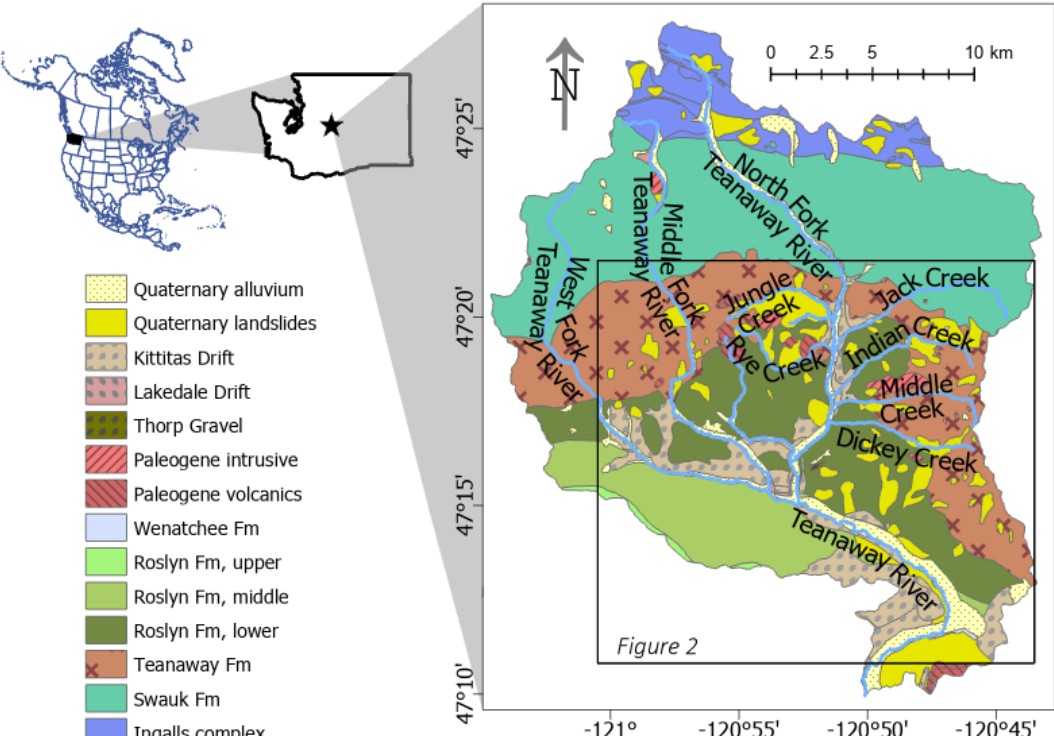

**Figure 1. Geologic map of the Teanaway watershed (Tabor et al., 1982). Upper left inset shows location of Washington State in North America, and the location of the study area (star) within Washington State. Box shows location of Figure 2.**



## 2 Background

### 2.1 Study site

The Teanaway River is located in central Washington State, northwest USA, four miles east of Cle Elum, WA (Figure 1). This
single-thread river has three main tributaries known as the West Fork, the Middle Fork, and the North Fork which all flow into
the main Teanaway River about 10 miles upstream of its confluence with the Yakima River. The region receives between 980
and 1230 mm of precipitation annually and is typically snow-covered during the winter (U.S. Geological Survey, 2012) with
large fires occurring every 300-350 years (Agee, 1996, 1994), though high-intensity burns are limited to less than 1 km$^2$
(Wright and Agee, 2004). Mapped faults do not offset Quaternary alluvium and exhumation and Holocene denudation rates
are low at 0.05 mm/yr and 0.1 mm/yr respectively (Moon et al., 2011; Reiners et al., 2003). The branches of the Teanaway
River were splash dammed from 1892-1916 (Cle Elum Tribune, 1891; Kittitas County Centennial Committee, 1989) and the
*Pinus ponderosa* forests were logged from the 1890s through the 1940s.

The majority of rock units in the study area were deposited during the Eocene (Figure 1). The lower Eocene Swauk Formation,
composed of dark sandstone with small amounts of siltstone and conglomerate, unconformably overlies the Jurassic Ingalls
Complex and is ~4800 m thick (Tabor et al., 1984). The Swauk Formation is folded with dip directions generally to the south
(Tabor et al., 1982). The middle Eocene Teanaway Formation unconformably lies on the steeply tilted Swauk Formation. The
Teanaway Formation ranges from 10 to 2500 m thick and is composed of basaltic and andesitic lava flows interbedded with
tuff, breccia, and feldspathic sedimentary rocks (Tabor et al., 1984). Because of its resistance to weathering, this formation
forms most of the taller and more rugged peaks in the Teanaway area. Rhyolite flows from this formation have interbedded
with the conformable upper Eocene Roslyn Formation and outcrop through the study area as dikes (Tabor et al., 1984). The
youngest surficial rock unit, the Roslyn Formation, covers most of the lower-elevation and landslide-prone study area. The
unweathered white and weathered yellow immature sandstones were deposited conformably on the Teanaway Formation in
the late Eocene (Tabor et al., 1984). The Roslyn and Teanaway formations lie relatively flat or gently tilted to the southwest
compared to the Swauk Formation, and are very susceptible to erosion and sliding due to the interbedded tuffs, paleosols,
clays, and silts (NFTWA, 1996; Tabor et al., 1982).

Overlying the Eocene units are glacial and mass wasting deposits. Glacial terraces originate from the Thorp and Kittitas
glaciations at 600 ky and 120 ky, respectively (Porter, 1976). During drift deposition, glaciers from the Cle Elum catchment
to the west overtopped the dividing ridge and entered the West Fork and lower Middle and North Fork Teanaway valleys.
Thorp and Kittitas moraines, composed of poorly sorted gravels and cobbles, are present at the eastern edge of the study area
near the outlet of the mainstem Teanaway into the Yakima River and on the ridges surrounding the West Fork Teanaway
(Porter, 1976). The Thorp drift sediments are heavily eroded and therefore less visible than the Kittitas drift sediment, which
has been modified by mass wasting (Porter, 1976).






Geologic mapping has identified several Quaternary mass wasting deposits in the Roslyn and Teanaway formations (Figure 1) and subsequent reports have focused on landslides near stream banks (NFTWA, 1996). Landslides are as old as late Pliocene and are concentrated near rhyolite tuffs and a weathered surface in the Teanaway Formation, which form planes of weakness (NFTWA, 1996). Although closed depressions and ponds are visible in the lidar and suggest some recent activity, landslides

are not easily distinguished in aerial photography or in the field. Lidar in 2015 and 2018 (Quantum Spatial, 2018, 2015) revealed the extent of these slides, but no studies since have quantified landslide volumes or constrained the timing or mechanism of sliding.

## 2.2 Habitat disturbances in the Teanaway

Current salmonids in the Teanaway include three species of *Oncorhynchus* (*mykiss*, *clarki*, and *tshawytscha*), two species of

*Salvelinus* (*fontinalis* and *confluentus*) and *Prosopium williamsoni* (NFTWA, 1996). *O. mykiss* and *O. tshawytscha* were historically abundant in the upper Yakima River watershed, which includes the Teanaway (Bonneville Power Administration, 1996), but populations have declined in the upper tributaries since Roza Dam construction in 1940 and anthropogenic alteration in the late 19th century. Currently only a few migratory fish make it to the Teanaway basin each year (NFTWA, 1996). Since 1999, *O. tshawytscha* are released in the North Fork Teanaway near the junction with Jack Creek (Figure 1) as part of an effort

to restore salmon populations in the Yakima River basin (Pearsons and Temple, 2010).

The *Onchorhynchus* genus, which includes salmon and western trouts, experienced significant speciation between 20 and 6 Ma, coincident with mountain building along the Pacific Coast. The Teanaway basin, in the central Cascade Range, likely didn't begin to uplift to the present high elevations until 8 Ma, as indicated by thermochronology (Reiners et al., 2002) and

deformation of Miocene Columbia River Basalts (Mackin and Cary, 1965). Tectonic forces continued to dominate the disturbance history until the Pleistocene, when glaciation became the main disturbance. At least twice during the Pleistocene, at ~600 and 120 ky, valley-filling glaciers overtopped the ridge between the western Cle Elum drainage and the Teanaway (Porter, 1976). Evidenced by moraine deposits and 35 m high glacial terraces (Washington Division of Geology and Earth Resources, 2016), these glaciers blocked the entirety of the West Fork and mainstem Teanaway and the lower ~10 km of the

Middle and North Fork Teanaway rivers. During the latest Pleistocene, megafloods from glacial Lake Missoula scoured the Columbia Plateau and would have killed any anadromous fish populations migrating between the Pacific Ocean and the Teanaway basin (Waples et al., 2008).

Since the end Pleistocene, disturbances to *Onchorhynchus* populations in the Teanaway have been driven by slow climate

changes and punctuated disturbances such as floods, wildfires, and landslides (Waples et al., 2008). Temperatures increased approximately 1.5°C by the mid-Holocene, and have slowly decreased 0.5°C to the late Holocene; at the same time, effective moisture was steady through the first half of the Holocene before rising rapidly in the last 5 ky (Shuman and Marsicek, 2016).





In addition to climate changes, isostatic rebound from the Last Glacial Maximum (LGM) continued to impact base level until ~5 ky (Beechie et al., 2001). Thus, it's likely that Teanaway *Onchorhynchus* populations stabilized by approximately 4-5 ky.


Since 4-5 ky, punctuated disturbances, including anthropogenic alteration, have episodically affected habitat in the Teanaway. The Teanaway basin experiences large fires every 300-350 years, though high severity burn areas with high vegetation loss tend to be less than 1 km$^2$ (Wright and Agee, 2004). Fires increase erosion and surface runoff to streams, decreasing habitat for eggs and fry but increasing habitat for juveniles and adults through delivery of large wood (Flitcroft et al., 2016). Similarly,

episodic flooding and regional droughts could have affected bed scour, aggradation, and stream flow and altered habitat for a period of years to decades (Waples et al., 2008). In the last 150 years, road construction and deforestation of the hillslopes resulted in greater runoff and erosion (NFTWA, 1996), and splash-dam timber transportation scoured the West and Middle Fork Teanaway rivers to bedrock (Schanz et al., 2019) and removed spawning habitat.

While the fire and anthropogenic history of the Teanaway have been quantified (e.g., Wright and Agee, 2004; Schanz et al., 2019), the landslide history remains unclear with very few temporal constraints on landslide activity—noted as late Pliocene and younger (Tabor et al., 1984)—and only rough spatial constraints on landslide locations and types (Figure 1, Washington Division of Geology and Earth Resources, 2016). Here, we use the recently collected lidar (Quantum Spatial, 2018, 2015) of the Teanaway basin to comprehensively map landslides and develop a surface-roughness curve for the basin that allows us to

analyze earthflow distributions through the Holocene, with implications for in-stream *Onchorhynchus* habitat.

## 3 Methods

Our analysis focuses on the entirety of the Teanaway basin, though the majority of the earthflows are found within tributaries to the North Fork Teanaway River. To identify the temporal and spatial distribution of earthflows, we use geomorphic mapping in conjunction with a directional roughness metric to identify and relatively date earthflows in the Teanaway basin. Other

studies (e.g., Mackey and Roering, 2011) use tree and object tracking to measure earthflow velocity; we attempted to do this but found the dense vegetation and high tree growth rates prevented us from accurately matching objects between image pairs. Thus, we rely on surface roughness to give relative earthflow activity.

### 3.1 Earthflow mapping and absolute dating

We first created a detailed earthflow map for the study region. All visually-identifiable landslides within the North Fork

Teanaway area were mapped in ArcGIS from one-meter resolution lidar (Quantum Spatial, 2018, 2015) at a scale of 1:5000. Earthflows were classified from this dataset based on: hourglass shape, narrow width and long length of slide zone, visible levees or shear zones at the edges, and flow-like morphologies (Baum et al., 2003; Nereson and Finnegan, 2018). These morphologic clues degrade over time and bias our earthflow mapping to younger slides; however, we focus our analysis on



Holocene earthflow activity to minimize this bias. In addition to earthflows, we also identified translational slides—noted as
thin deformation over a wide area—and rotational slides with clear rotated headscarps and toes.

We dated select earthflows using buried charcoal found within the earthflow toe deposits. Sampled earthflows were selected
based on a visual estimate of roughness and potential for a fresh exposure via road or stream erosion. In the field, we removed
10-50 cm of material from the toes of earthflows exposed by stream cuts or roadcuts to find 2-5 grams of charcoal. We collected
radiocarbon samples from six different earthflows (Table 1). The samples were sent to the Center for Applied Isotope Studies
(CAIS) lab at the University of Georgia and were dated using Accelerated Mass Spectrometry (AMS); ages were calibrated to
calendar years using Intcal20 (Reimer et al., 2020).

In three cases where earthflows dammed the valley and formed lakes, we estimate the onset of valley blockage and an
approximate earthflow age by using the sedimentation age of the lake. We estimate a pre-earthflow valley bottom using the
techniques in Struble et al. (2020) and subtract this from the lidar surface elevation to give an estimate of the sedimentation
volume post-earthflow. We use a nearby mid-Holocene denudation rate of 0.1 mm/yr (Moon et al., 2011) combined with the
upstream contributing drainage area to calculate the sedimentation age and use this as an estimate of the earthflow age.

**3.2 Flow directional surface roughness and relative ages**

To relatively date the earthflows, we created a surface roughness age calibration model similar to that used to date rotational
slides in Washington State (LaHusen et al., 2016). Earthflows start with a unidirectional flow morphology and gradually diffuse
to less directional roughness, in contrast to rotational slides which start with uneven roughness in all directions. To account
for the unique flow morphology of earthflows, we used a flow directional Median Absolute Differences (MAD) index
(Trevisani and Rocca, 2015). MAD is a bivariate geostatistical index that analyzes Digital Elevation Models (DEMs) on
multiple dimensions (Trevisani and Cavalli, 2016), giving us a directional roughness index for each one meter raster cell across
the study area. This directional roughness is combined with flow directions derived from the DEM to analyze surface roughness
relative to flow direction (Trevisani and Cavalli, 2016) in which a high MAD value represents very directional regions, while
a low MAD represents relatively planar regions.

We first tested the relationship between MAD and earthflow age by extracting elevations from an earthflow along Jungle
Creek where we obtained radiocarbon sample 8-1-20-1 (Figure 2, Table 1). We chose this earthflow because it has clear flow
lines and blocks the majority of the stream valley with an outlet eroded through. This suggests the earthflow has been active
recently, to block the valley, yet is not so strongly active that the stream is permanently dammed. We applied two-dimensional
diffusion to the earthflow surface, based on Eq (1):

$\frac{dz}{dt} = -K\frac{dz^2}{d^2x}$,                                                               (1)





where dz is change in elevation, dt is the timestep, and dx is the spatial resolution. The diffusion rate, K, is estimated as 0.002 $m^2$/yr based on regions in a similar climate (Martin, 2000). We ran the diffusion model for 10 ky and calculated MAD using the steps below every 2 ky.

MAD is calculated using the residual roughness; we first smoothed the one-meter lidar-derived DEM over a 3x3 window followed by a 5x5 window (Trevisani and Cavalli, 2016) and subtracted the smoothed DEM from the original DEM to obtain a residual DEM of roughness elements. The MAD index (https://github.com/cageo/Trevisani-2015) was run with this residual DEM and calculated the directional roughness over an 8 m radius window. We chose this window so that we examine a similar spatial scale as the 15x15 window used by LaHusen et al. (2016). We calculated flow direction across the smoothed DEM and
created a raster with the MAD values in the direction of flow for each cell. Finally, we used Focal Statistics to calculate the standard deviation of the directional roughness (MADstd) for each earthflow; from our diffusion model simulations, MADstd had the highest correlation with age ($R^2 = 0.99$).

### 3.3 Valley width

To examine the influence of landslides on habitat, we measured valley width along the tributaries of the North Fork Teanaway.
The mainstem and three forks of the Teanaway all have wide valleys that are unaffected by earthflows. In contrast, the tributaries of the North Fork contain earthflow dammed-lakes. We extracted valley width from Jungle, Rye, Dickey, Middle, Indian, Jack, and an unnamed creek (Figure 1) by defining the valley floor as being less than 5% slope. We used an automated process in ArcGIS to extract a valley centerline, create transects every 100 m, and measure valley width.

## 4 Results

### 4.1 Landslide mapping

We mapped 363 landslides in the lower Teanaway basin (Figure 2). Earthflows compose 52% (187) of the mapped slides but made up 72.5% by area. Translational slides, which often had flow-like morphologies but were wider than they were narrow, were the second most frequent slide type with 39% (141) of the count and 16% of the area. We only identified 34 rotational slides, but several large complexes in glacial drift contributed a large total slide areal fraction of 10.7%.


Mapped landslides are mostly all north of the Main Fork Teanaway River, with the exception of 18 small landslides south of the Main Fork. The southern edge of the landslide area appears to be bound by the extent of Pleistocene glaciation (Figure 1); perhaps glaciation removed pre-existing landslides or the muted topography from glacial erosion is less prone to mass movement. To the north, the landslide domain is bound by the start of the Swauk Formation, which has little to no mappable
landslides in it.

Earth **Surface**
**Dynamics**
Discussions
EGU

We extracted slope and aspect for each landslide. The slope distribution, measured based on the smoothed one-meter lidar, was similar for all three slide types (Figure 3), with modal slopes of 10 to 15 degrees for earthflows and rotational slides, and slightly higher modal slopes in translational slides. For comparison, intact hillslopes in the Lower and Middle Roslyn

Formations have similar median slopes as the landslides, though the Lower Roslyn has slightly lower modal slopes than the three slide types. The average landslide aspect shows strong differences between earthflows, translational slides, and rotational slides (Figure 4). Fewer rotational slides were mapped, through there is a slight preference for south facing hillsides. Translational slides were more frequent on hillslopes facing the southeast quadrant as well as slopes facing north. In contrast, earthflows are most common with aspects facing the southwest quadrant (40.6% of earthflows) although the southeast quadrant

was also common (30%).

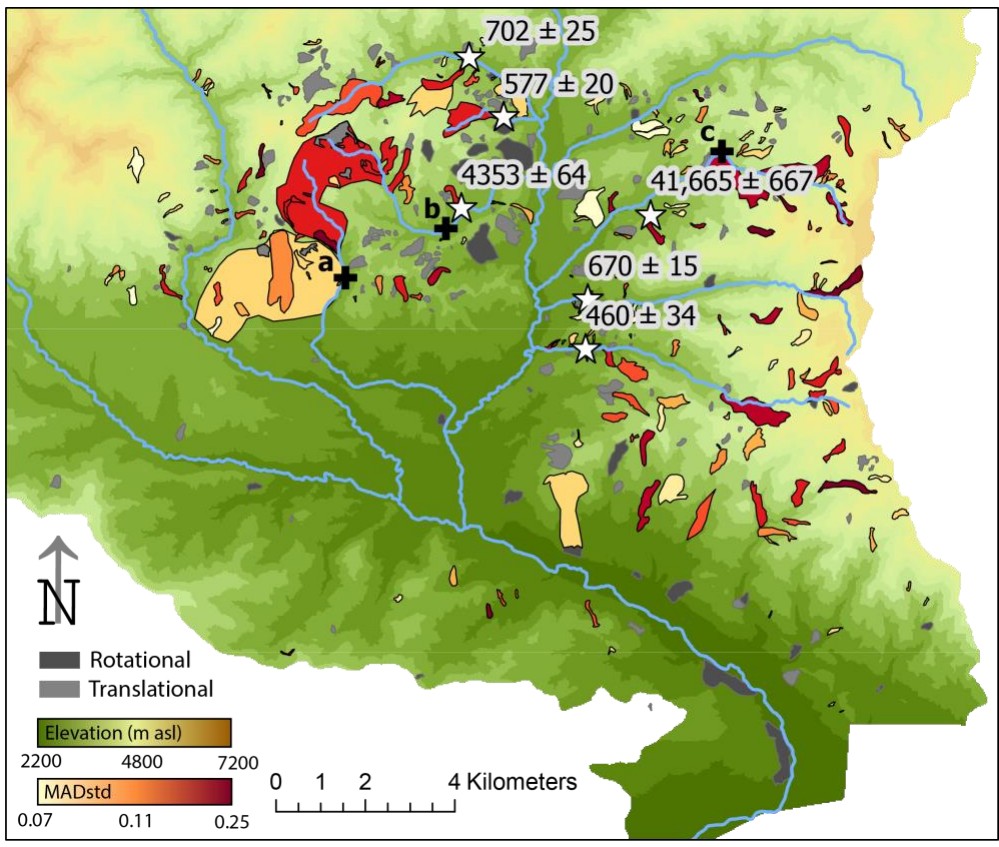

**Figure 2. Landslides mapped in the study area; rotational and translational slides are grey while earthflows are colored by their MADstd value. Radiocarbon locations and dates, in calibrated yr BP, are shown with white stars. Black crosses indicate locations of earthflow-dammed lakes where sedimentation ages are derived: a – unnamed creek; b – Rye Creek; and c – Indian Creek. Extent**
**of region is shown in Figure 1. Background elevation data from Quantum Spatial (2015; 2018).**





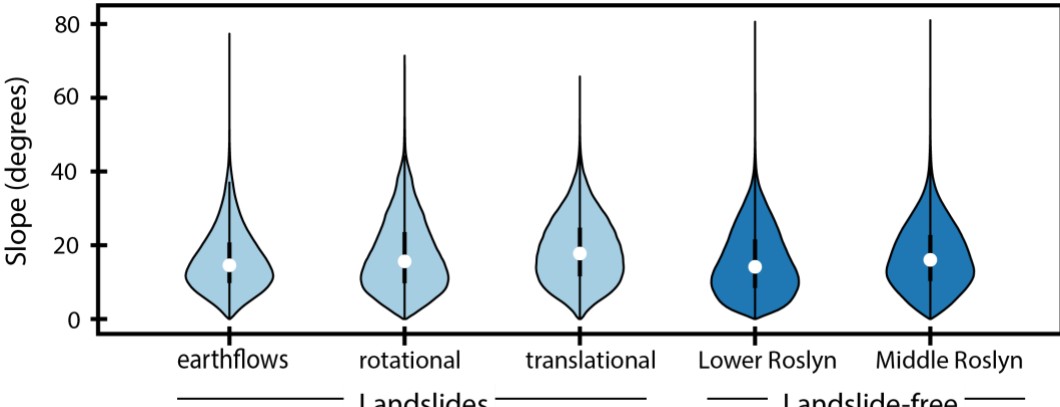

**Figure 3. Distribution of slopes across landslides and landslide-free topography. White dot shows median values, with first and third quartiles shown by the thick black lines.**

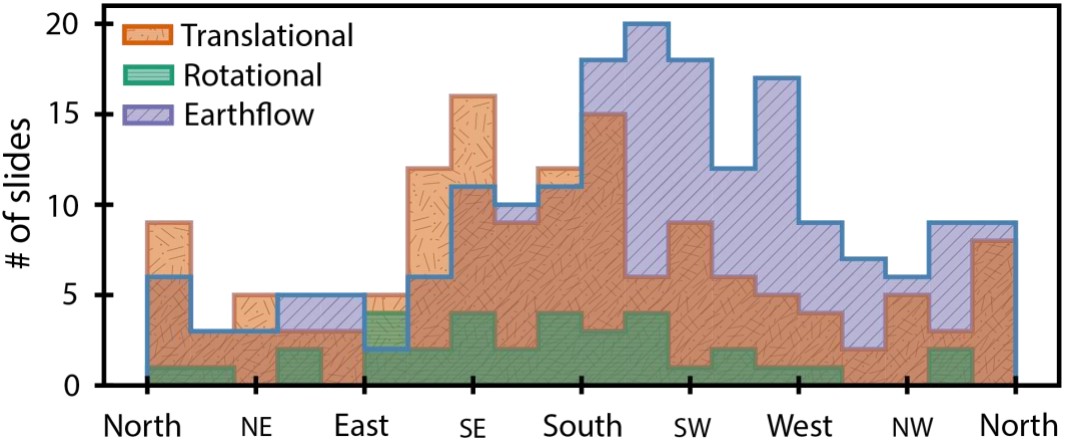

**Figure 4. Average landslide aspect, binned by 18 degrees, for mapped translational slides, rotational slides, and earthflows.**

### 4.2 Absolute earthflow ages

Age results from radiocarbon dating range from 370 to 36,750 carbon-14 years before present, or $460 \pm 34$ to $41,665 \pm 237$ calibrated years before present (yr BP) (Table 1). Samples were taken from the toe of earthflows, and represent charcoal that was originally deposited in regolith then transported through earthflow movement. Thus, the age given by radiocarbon dating

is a measure of 1) the inherited age of the charcoal, 2) regolith development, 3) earthflow transport, and 4) deposition at the earthflow toe. We cannot use our ages to directly date the last earthflow activity, but it does provide a maximum estimate of the most recent earthflow activity. Four of the six dated slides have ages less than 1 ky, and indicate earthflow activity has been frequent in the last 1 ky.

The ages we derived from sedimentation rates and lake volume are minimum ages, as all three earthflow-dammed lakes are nearly completely filled with an outlet carved through the damming earthflow. Based on an average denudation rate of 0.1


Earth **Surface**
**Dynamics**
Discussions

mm/yr, the lake formed along Indian Creek (Figure 2) took approximately 453 years to fill to the current level, indicating the earthflow has been constricting Indian Creek for at least that long. The lake along Rye Creek, formed just upstream of earthflow carbon site 8-3-20-3, took 346 years to fill with sediment to the modern level, and the lake along the unnamed creek took

approximately 270 years to fill. The Rye Creek earthflow was dated with charcoal to 4353 yr BP, and so the true earthflow age is likely somewhere between 346 and 4353 years.

Table 1. Radiocarbon dates

| Lab ID | Tributary name | Latitude | Longitude | C-14 yrs BP (2 sigma) | calibrated yr BP (2 sigma) | MAD std |
|---|---|---|---|---|---|---|
| 8-3-20-1 | Jungle Creek | 47.34689 | -120.87804 | 790 ± 20 | 702 ± 25 | 0.142 |
| 8-3-20-2 | unnamed tributary to Jungle Creek | 47.33463 | -120.87036 | 640 ± 20 | 577 ± 20 (p = 0.57) | 0.138 |
| | | | | | 643 ± 18 (p = 0.43) | |
| 8-3-20-3 | Rye Creek | 47.31456 | -120.87959 | 3910 ± 20 | 4353 ± 64 | 0.141 |
| 8-3-20-4 | Middle Creek | 47.29731 | -120.84273 | 730 ± 20 | 670 ± 15 | 0.116 |
| 8-4-20-1 | Indian Creek | 47.31481 | -120.82517 | 36750 ± 20 | 41665 ± 237 | 0.146 |
| 8-4-20-3 | Dickey Creek | 47.28752 | -120.84302 | 370 ± 20 | 460 ± 34 (p = 0.61) | 0.094 |
| | | | | | 349 ± 29 (p = 0.39) | |

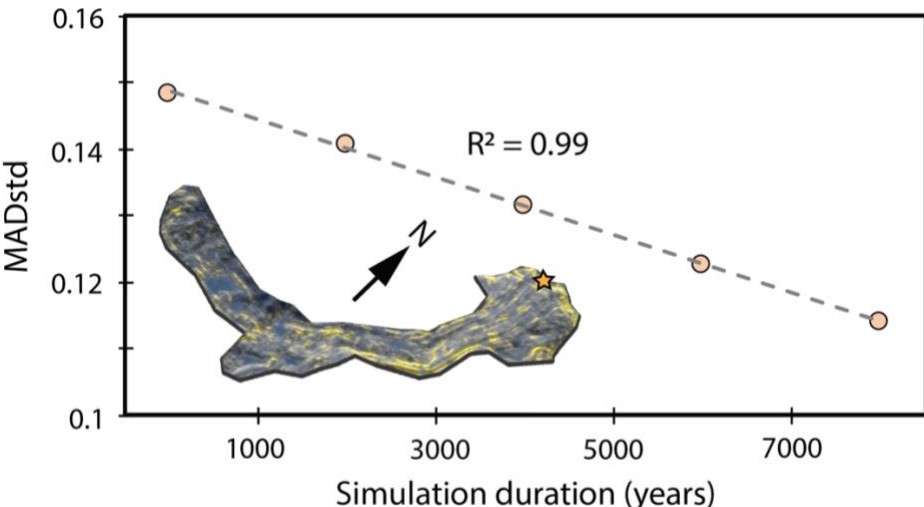

**Figure 5. MADstd values for simulated diffusion across the Jungle Creek earthflow. Inset images shows the Jungle Creek slide with modern (simulation time = 0) MAD values where yellow are high directional MAD and blue are low. Star shows location of sample 8-3-20-1.**





**Figure 6. Distribution of MADstd values by subbasin within the Teanaway in A-K. Location of each subbasin by color and extent of mapped earthflows is shown in L.**

### 4.3 Relative earthflow ages

Simulated diffusion across the Jungle Creek earthflow shows a strong linear relationship between MADstd and earthflow age (Figure 5), suggesting that as earthflows stop moving, the directional roughness becomes more similar across the surface as diffusion creates a more multi-directional surface. Although the flow features on the earthflow are linear, giving similar MAD





values, orthogonal flow off the flow features and scarps creates a highly variable MAD and thus a high MADstd. While our
simulation gave an equation relating age and MADstd, we do not apply this equation to the study area because: 1) we do not
know the site-specific diffusion rate and 2) we do not know how the diffusion rate changed over the late Quaternary. However,
we can assume that the diffusion rate and associated variations are similar across our study area, where climatic and biotic
forcings are relatively uniform and lithology is mostly Roslyn Formation. This allows us to create a relative map of earthflow
activity based on the MADstd (Figure 2).

The majority (122 or 65%) of the earthflows are located in the North Fork Teanaway compared to 24 in the Middle Fork, 5 in
the West Fork, and 37 in the mainstem. Earthflows in the mainstem cluster around MADstd values of 0.13 and lower, with a
few slides near 0.18 or greater (Figure 6). This suggests most earthflows in the mainstem Teanaway basin are older, though
we can't assign an absolute age. Similar age patterns are seen for Jack Creek, Middle Creek, Dickey Creek, Jungle Creek, the
Middle Fork, and the North Fork excluding tributaries (Figure 6A-D, G-I), implying most of the earthflows in these sub-basins
are older with a few younger, potentially active earthflows. In contrast, the unnamed tributary, Indian Creek, and Rye Creek
tend to have more earthflows at a higher MADstd value implying active, or at least more recent, earthflow movement than the
surrounding basins.


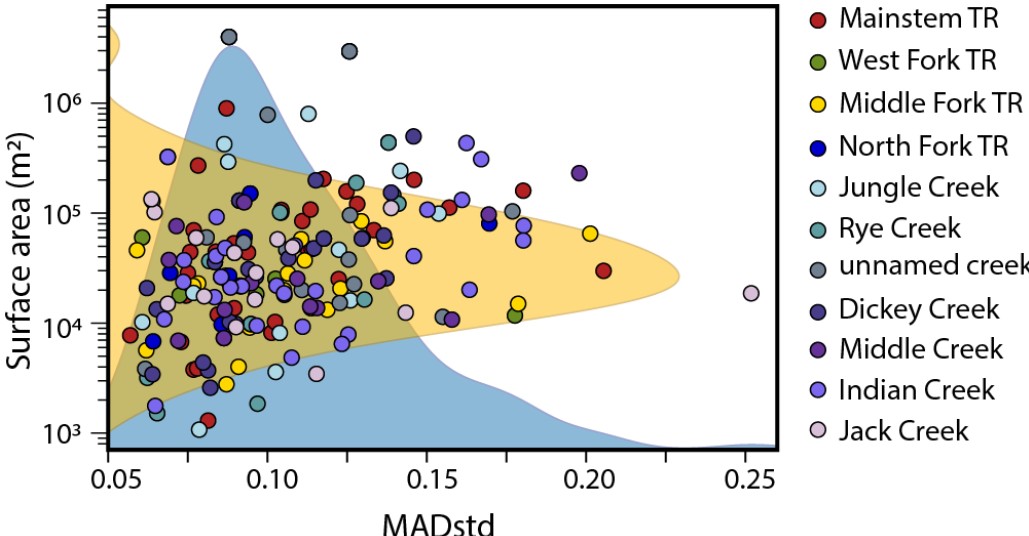

**Figure 7. MADstd values compared against earthflow surface area for the basins in Figure 6L. Curves in the background show the kernel density probability function for surface area (yellow) and MADstd (blue). In the legend, TR = Teanaway River. The North Fork TR excludes all the sub-basins listed below it in the legend.**





## 4.4 Earthflow area

We calculated earthflow area for each slide in ArcGIS and compared the area against the MADstd measure of relative age (Figure 7). Slide area ranges across three orders of magnitude, though most slides are on the order of $10^4$ m$^2$. Two large earthflow complexes of 3-4 km$^2$ lie outside the log-normal distribution of slide sizes. There is only a very low correlation between MADstd and slide area, with an $R^2$ of 0.0007, suggesting there is no significant link between slide activity and slide size.

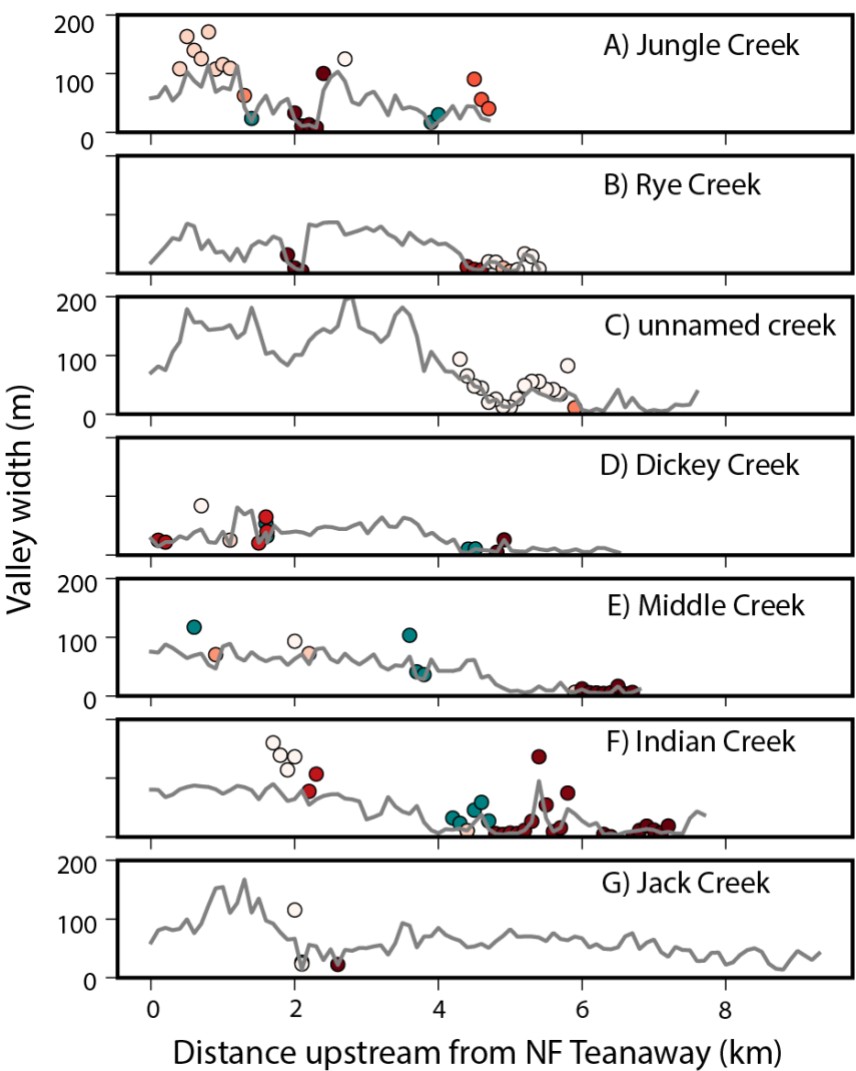

**Figure 8. Valley width of the North Fork tributaries, arranged counterclockwise from the northwest. Circles show the locations of earthflows; circle offset from the valley width line reflects the distance between the earthflow toe and the valley centerline. Colors indicate MADstd relative to earthflows within the tributary in which red are high MADstd and light pink are low MADstd. Teal circles are translational or rotational slides.**



## 4.5 Valley width

Valley width generally decreases upstream for the seven tributaries we examined, although the decrease is not consistent (Figure 8). Jungle Creek has its narrowest width, equivalent to the channel width, halfway up the valley where a high MADstd earthflow pinches the valley. The valley width immediately upstream is 100 m wide, comparable to the widest part of the valley at the mouth of Jungle Creek. Similarly, Rye Creek's valley is pinched to the channel width at 2 km upstream and widens immediately upstream to the widest values noted along the tributary. Similar trends of narrowed valleys with wider sections immediately upstream are seen in the other tributaries, though the trends are less strong. Rye, Middle, Indian, and the unnamed creek are confined by earthflows in the upper 1-2 km; these earthflows form the valley walls and bottom and constrain the valley width to the active channel width. Unsurprisingly, when the earthflow toe is further from the valley centerline, there is less of an impact on the valley width; however, the earthflow may exert a lingering influence as seen by the variations in valley width and lack of a uniform downstream widening trend in the lower 2 km of Jungle Creek that is bound by an earthflow complex (Figure 8A).

## 5 Discussion

### 5.1 Verification of MADstd relative dating

We apply the MAD surface roughness metric (Trevisani and Rocca, 2015) to earthflows in order to relatively date them; our numerical simulation shows that MADstd strongly correlates with relative earthflow age (Figure 5) and topographic relationships support the relative dating technique. For instance, active earthflows are more likely to block tributary valleys in contrast to older, less active earthflows whose deposits can be eroded by the stream to re-form a wide valley. In our analysis of valley width, higher MADstd earthflows were associated with narrower valleys, often narrowed to the active channel width (Figure 8). An outlier to this is the earthflow from 4-6 kilometers along Rye Creek and the unnamed creek (Figure 8B, C) with a low MADstd but strong effect on valley width. Both of these earthflows are large earthflow complexes (3-4 km$^2$) and the MADstd value of the entire complex may not represent the locally active portions that affect the two creeks.

Relative dating by MADstd is further supported by the cross-cutting relationships between earthflows, indicating that it can be applied between slides with some limitations. In our study area, there are 22 instances of earthflows clearly overlapping with another, in which morphologic clues can be used to relatively date them. Of these, 15 had MADstd values that reflected the cross-cutting relationship. In cases where the MADstd gave incorrect relative ages, five were on earthflow complexes. MADstd appears to not work as well across large earthflow complexes where there is more heterogeneity in activity and less defined flow lines and scarps. If we disregard earthflow complexes, then only two of 20 cross-cutting relationships are not reflected by the relative MADstd values.





Finally, our lake sedimentation ages placed the valley-blocking landslides along Indian, Rye, and the unnamed creek in a rough chronologic order that can be compared to the MADstd relative ages. That the impounded lakes are filled but still preserved

indicates the earthflows damming the valley must be relatively active, otherwise the channel would carve a stable new valley through the earthflow deposit and start to incise the lake fill. The sedimentation ages support this inference, showing the earthflows have been active in the last few hundred years. Recent activity is also supported by the relative MADstd values in each tributary; the MADstd values for the Rye Creek and Indian Creek lake-forming earthflows are at the higher end of calculated MADstd values for those tributary valleys at 0.14 and 0.16, respectively (Figure 6). However, the MADstd value

for the earthflow complex creating the lake along the unnamed creek is 0.087 and is on the lower end of MADstd values. But, as we discussed above, we have lower confidence in the applicability of MADstd to earthflow complexes and believe the earthflow-wide averaging over a 4 km$^2$ slide is unreliable. When we apply a 5 m moving window MADstd to the earthflow complex along the unnamed creek, a section with higher MADstd is apparent at the base of the lake and may indicate a more active section of the complex is responsible for the valley damming and lake formation.


These observations give us confidence that MADstd can be applied to relatively date earthflows. The original flow directional MAD metric picks up flow features such as scarps, debris flows, and channels that are missed by isotropic roughness metrics (Trevisani and Cavalli, 2016). In the case of earthflows, high and low flow directional MAD values are associated with the strong lineations; as flow follows the crests and hollows, the >1 m lineations also direct flow orthogonal to crests (Figure 5).

By taking the standard deviation, we can highlight the parallel and orthogonal flow that is characteristic of >1 m scale lineations; however, it is important to note that this method would not work if the elevation model resolution is greater than the lineation scale. Compared to other metrics applied to landslides, the MADstd includes a flow directional roughness and detrends the data, both of which have been found to improve landslide identification accuracy (Berti et al., 2013; McKean and Roering, 2004). Previously used surface roughness metrics often have trouble capturing the top of earthflows and

differentiating between rough, forested terrain and landslide roughness (Berti et al., 2013). When the MADstd is calculated over a moving 5 m radius window, rather than over a single earthflow, forested hillslopes are clearly delineated from earthflows (Figure 9). The roughness elements from trees are isotropic and give MADstd values near zero. The scarp, flowlines, and toe produce strong lineations in the landscape that light up in the MADstd plots, due to the parallel and orthogonal flow over the 1 m DEM. Even smaller earthflows, such as that in the center of Figure 9B that is 3600 m$^2$, are identified with the 5 m moving

window MADstd. This advantage over previous, isotropic methods of calculating surface roughness and identifying landslides indicates MADstd is an appropriate method for use in identifying and mapping earthflows, though we caution that the DEM resolution size must be less than the scale of earthflow lineations.



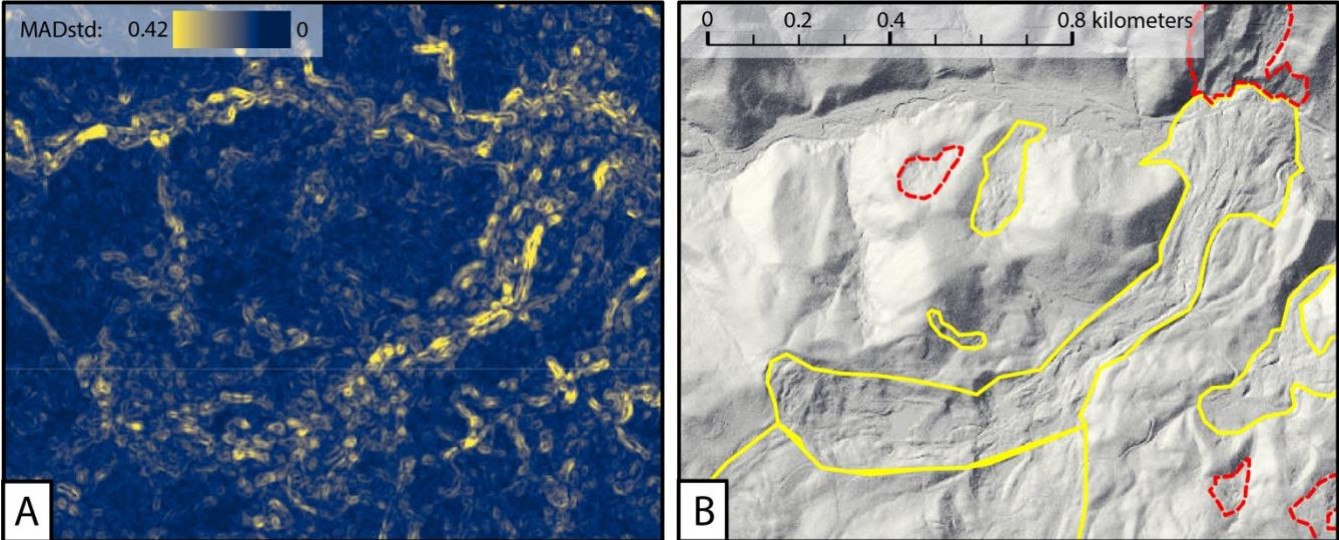

**Figure 9. Comparison of MADstd on earthflows and intact hillslopes. A) MADstd calculated in a moving 5m radius window for a section of the study area near Jungle Creek. B) The same area with a 1 m shaded relief image (Quantum Spatial, 2015) and mapped earthflows (yellow) and translational slides (dashed red). The area is forested with dense *Pinus ponderosa* stands with a minimum density of 0.7 trees per 100 m$^2$.**

**5.2 Drivers of earthflow motion**

Our aspect analysis showed a strong preference for earthflows to be oriented towards the southwest quadrangle (Figure 4), and we hypothesize that this reflects a bedding plane control on earthflow location. The Roslyn Formation and the Teanaway Formation are gently dipping to the southwest with dip angles ranging from 10 to 30 degrees (Tabor et al., 1982), comparable to the modal and median earthflow slopes (Figure 3). There is some variability in the bedding orientation as the Teanaway and Roslyn formations curve to the west, but only 8.5 percent of earthflows by area are located in this southeast-dipping region. Since the Roslyn and Teanaway are conformable, the bedding plane orientation also reflects the mid-Eocene landscape surface, and thus the orientation of paleosols within the two units. Previous work has noted that paleosols and volcanic flows interspersed in the Teanaway and Roslyn formations form planes of weakness for landslides (NFTWA, 1996). That our observed slopes and aspects match the bedding orientation supports this finding and indicates the bedding provides a first-hand control on the orientation of earthflows in the Teanaway basin.

While bedrock orientation may control the locations and orientations of earthflows in the Teanaway basin, the timing of earthflow activity is likely set by climate. The combination of radiocarbon ages, sedimentation ages, and the relative MADstd distributions suggests most earthflows were active in the mid or late Holocene. Our radiocarbon ages are mostly less than 1 ky, as are the three sedimentation ages, and this indicates earthflow activity during this time to transport charcoal to the slide toes and to block valley bottoms. If we assume that we can roughly relate the <1 ky age and the corresponding MADstd values of 0.1 to 0.14 of the dated slides (Table 1), then we predict that earthflows with MADstd of 0.1 or greater were active in the



late Holocene. For many of the North Fork tributaries, this MADstd range represents a frequency peak (Figure 6), showing an increase in earthflow activity in the late Holocene. Another frequency peak of MADstd values from 0.05 to 0.10 represents an older period of earthflow activity; using the decrease in MADstd with age noted in our simulation (Figure 5), this peak is likely a few thousand years older than the late Holocene spike, and can roughly be attributed to the mid-Holocene; this estimate is
supported by climate records as well, as we discuss in the following paragraphs.

We hypothesize that the timing of earthflow activity in the Teanaway basin is controlled by climate, which primed the landscape through regolith production in the late Pleistocene and early Holocene, and triggered earthflows during the warm and wet mid to late Holocene. During the Last Glacial Maximum (LGM), which ended 11.8 ky, the climate was cold and dry
(Riedel, 2017); while the Teanaway basin was not directly covered in ice during this time, it was subjected to frost cracking (Marshall et al., 2021). Frost cracking can cause a 2.5x increase in catchment erosion rates, as documented at Little Lake, OR—a site in similar lithology but 400 km further from ice sheets (Marshall et al., 2015). In the Teanaway basin, regolith production likely increased during this time, though the dry and unglaciated conditions prevented this regolith from being eroded. After the end of the LGM, the Teanaway basin warmed rapidly until the mid-Holocene—around 6 ky—but stayed dry
(Shuman and Marsicek, 2016). This dry and warm period likely caused desiccation cracks to form in the soil and, combined with the LGM dry and cold conditions, could have primed the landscape for earthflows (McSaveney and Griffiths, 1987).

After 5 ky, effective moisture rose rapidly and continued to rise (Shuman and Marsicek, 2016), triggering earthflow motion. The deep regolith produced during the LGM supplied material for earthflows, while desiccation and frost cracks created
conduits for rainfall to reach deep in the regolith and form a sliding surface on bedding planes (McSaveney and Griffiths, 1987). Earthflow MADstd values, when separated by subbasin, often show an early peak around 0.05 to 0.10 (Figure 6) that could represent earthflows that initiated during the mid-Holocene warm and wet period. However, regolith production is unlikely to continue to keep pace with earthflow motion (Mackey and Roering, 2011), and earthflow motion would have slowed or ceased. Local conditions, such as the degree of hillslope hollowing, precipitation, fires, and local volcanic interbeds
and paleosol surfaces in the Roslyn and Teanaway formations (NFTWA, 1996) could increase regolith production and lead to sustained earthflow activity through the late Holocene for a few slides. Human modification since 1890, which included deforestation and road building, may have further altered the rate of water infiltration and allowed earthflow reactivation.

**5.3 Habitat disturbance**

Only a small proportion of *Onchorhynchus* habitat is in direct contact with an earthflow. All of the North Fork tributaries
examined in Figure 8 initiate on an earthflow or earthflow complex, with the exception of Jack and Dickey creeks; however, these earthflows are in headwater streams too narrow and steep to host summer habitat for *Onchorhynchus* (NFTWA, 1996). In the Teanaway forks and mainstem, no earthflows impinge on the valley width. Only a relatively small number (10 of 187) of mapped earthflows in the North Fork tributaries are in direct contact with in-stream habitat for non-migratory and juvenile





*Onchorhynchus*. These earthflows range in size from a large earthflow complex of 4 km$^2$ to smaller flows of 14,000 m$^2$ and
show mostly late Holocene activity; nine have MADstd values in the range of our estimated late Holocene activity or have
radiocarbon or sedimentation ages in the late Holocene.

The estimated earthflow ages for the Teanaway basin of mid- to late-Holocene indicate that earthflows were active at the same
time as *Onchorhynchus* habitat, disturbed by lingering LGM influences, was stabilizing (Beechie et al., 2001; Waples et al.,
2008). A handful of earthflows have continued to be active through the late Holocene and actively impact in-stream habitat
through channel narrowing and sediment delivery. Although earthflows can inhibit habitat development by introduction of
fine sediment, either the fines were eroded away or were supplied at a low rate, as fish were still able to spawn in large numbers
during peak earthflow activity (Bonneville Power Administration, 1996). Floodplain habitat was enhanced through earthflow
activity in Jungle, Rye and Dickey creeks, where valley widths are abnormally wide just upstream of earthflows (Figure 8).
Floodplain habitat is reduced where the earthflow pinches the valley, but the extent of widened valleys is much larger and
gives a net gain in habitat from earthflows.

Earthflows, fires, and human alteration are the largest disturbances to salmon habitat in the mid to late Holocene (Waples et
al., 2008), but changing climate poses a major concern for habitat moving forward. In the Teanaway basin, streamflow is
expected to decrease dramatically, due in part to a transition from snowmelt- to rainfall-driven hydrology. Low flows are
expected to decrease by 30-75% in the next decade, and high flows by 10-50% (Beechie et al., 2013). Maximum weekly water
temperatures will increase by 5-6 times, likely exacerbated by the shallow flows. In order to create resilient habitat for salmon,
floodplain habitat needs to be increased and in-stream pools and cold-water refuges created. Earthflows have helped build
some resiliency, particularly in Jungle, Rye, and Dickey creeks where valleys have widened upstream of earthflows and created
greater floodplain habitat (Figure 8). In summers 2019 and 2020, large wood jams were installed along the North Fork
Teanaway in an effort to increase channel-floodplain connectivity, create pools, and restore *Onchorhynchus* habitat (Mid-
Columbia Fisheries Enhancement Group, 2020). As managers continue to restore the habitat and protect it against climate
change, it will be important to continue to monitor activity along the ten earthflows directly abutting habitat, and to take
advantage of the widened floodplain habitat created by earthflows.

**6 Conclusion**

To examine the influence of earthflows on salmon habitat disturbance in the Teanaway basin, we mapped and dated earthflows
using 1 m lidar and a new relative dating method. The MADstd metric appears well-suited to identifying and relatively dating
earthflows, as it picks up flow directional variations in roughness and is able to ignore the influence of dense vegetation on the
elevation model. This is particularly useful for densely vegetated areas, where other roughness metrics have difficulty and
where object tracking is problematic to apply. In addition to MADstd relative ages, we used radiocarbon and sedimentation

ages to provide a few constraining absolute ages; these ages indicate that earthflows in the Teanaway basin were mostly active in the mid Holocene with some continued action through the late Holocene and present. Bedrock dip angle and orientation control the slide locations, with slide aspect and slopes roughly following the orientation of weathering and bedding planes. Climate controlled the timing of earthflows; late Pleistocene and early Holocene dry conditions built a deep regolith and 475 created conduits for water. Wetter conditions starting in the mid-Holocene triggered the earthflows. The earthflow activity coincides with stabilization of salmon habitat and documented *Onchorhynchus* abundance in the watershed, suggesting the earthflows did not negatively affect habitat. However, valley-blocking earthflows led to widened valleys upstream that can potentially help buffer *Onchorhynchus* populations against climate-changed induced low flows and higher temperatures over the coming century.

**Data and code availability**

The diffusion simulation code and input files can be access on https://github.com/schanzs/JungleCk_diffusion. Landslide information and dates are available at: https://doi.org/10.5281/zenodo.5153965.

**Author contribution**

SAS conceptualized the study, SAS and APC contributed equally to study design and methodology. APC and SAS acquired 485 funding for the study. SAS wrote the paper with contributions from APC.

**Competing interests**

The authors declare that they have no conflict of interest.

**Acknowledgements**

Funding to APC was provided by the Patricia J. Buster grant from the Colorado College Geology Department and radiocarbon 490 sample analysis was paid for by Colorado College. We thank Matt Cooney for GIS help, and Jamie and Catharine Colee for help in the field. Field work was conducted on the traditional territory of the Yakama and Wenatchi People.

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
