# Peer review of "Controls on earthflow formation in the Teanaway River basin, central"

_Earth Surface Dynamics, 2021_

## Author Comment (AC1)

Dear Referees,

Thank you for your constructive and critical comments on our submitted manuscript. We plan to address your comments through a major revision and will alter the manuscript focus to be on lithologic controls on earthflows, the effect of earthflows on valley width and sediment, and we will introduce/discuss MADstd as a potential method for relative dating earthflow activity. We believe these revisions will address your concerns, particularly regarding the relationship between MADstd and absolute age and regarding the historic analysis of salmon habitat. We are confident these revisions will improve the quality of the manuscript.

Our detailed comments are below.

Sincerely,
Sarah Schanz, on behalf of the authors

**Reply to Referee #1**

*RC1: First, to test the hypothesis that high earthflow frequency corresponds to times of large salmon populations, the salmon population would need to be independently known over the Holocene. I'm not a salmon expert, but that information doesn't seem readily attainable, and the manuscript doesn't provide any independent references that have determined it. Instead, there is a suggestion that salmon populations may have stabilized 4-5 ka based on very broad scale inferences about climatic and tectonic changes.*

AC: We agree that this hypothesis cannot be readily tested by the data, and plan to restructure the goals of the paper to focus on investigating earthflow activity in the Teanaway. The impact of earthflows on salmon habitat would then be relegated to an interesting point of future research in the discussion section, rather than the driving hypothesis.

*RC1: Second, the study did not present a significant, data-derived relationship between earthflow age and surface roughness, which would be necessary in order to use surface roughness to infer age of undated earthflows.*

*Age vs. activity. Since age (~time since a landslide happened) and activity (~how active the landslide is currently) both affect a landslide's surface roughness, it can be quite challenging to disentangle these two effects for earthflows in particular. This may be one reason why there is not a relationship between landslide age and roughness for the study site. Fig. 5 is a bit misleading on this point, since it shows a modeling result, which by design will smooth the landslide deposit over time. If you add the age and roughness values from Table 1 to the figure, it is clear that the radiocarbon-based ages and roughnesses are not correlated and do not follow the model-predicted trend. Those data are so scattered that that general inferences about mid or late Holocene timing aren't supported either (e.g. the smoothest landslide is the youngest (Dickey Cr.) and the oldest is the roughest (Indian Cr.)). Perhaps the fact that age and*

*roughness don't correlate is a useful observation for thinking about what other factors control surface roughness of these earthflows.*

*Landslide dates. As stated in the manuscript, charcoal provides only a maximum age, and often this age is not a close maximum. E.g. see Struble et al. (2020), GSA Bull. where charcoal from a landslide's deposit is up to thousands of years older than the landslide. Also in the present manuscript, the discrepancy between the minimum and maximum ages of the Rye Creek earthflow (346 and 4,353 yrs, respectively) suggests that either the earthflow's age isn't accurately enough known to determine an age roughness model, or that a single age for the earthflow may not be representative of its long term behavior, such as persistent movement and/or reactivations.*

AC: This is a great distinction, and thank you for the careful explanation. We assume that radiocarbon ages give some measure of the activity, but don't constrain when the landslide occurred (age). This is one reason we avoided directly comparing the radiocarbon ages to the MADstd values, noting that there is a wide variability in what the radiocarbon age represents relative to landslide age (and we plan to add directly the example in Struble et al. (2020)).

We plan to revise the manuscript to make it clear what is meant by age vs activity. For example, MADstd examines time since last activity, assuming the earthflow has stopped moving. This should not be compared directly to ages estimated from radiocarbon (maximum) but should be more closely related to minimum ages from sedimentation (minimum). We will make it clear that earthflows are not a single event, thus a single age, or even a single dating technique, will not provide complete information on earthflow age and activity.

We also plan to shift the focus of the manuscript away from salmon habitat and timing of earthflow activity. We recognize we did not have enough information to attempt this before. Instead, following Referee #2's suggestions, the manuscript will be edited to focus on lithologic controls on earthflows, impacts of earthflows on valley width and sediment, and introduction and discussion of a potential method to rate earthflow activity (but make it clear that we are not tying MADstd directly to absolute dates/ages). We believe this should remove many of Referee #1's concerns about the validity of using MADstd to infer ages and inferences about earthflow timing.

*RC1: An exciting, if somewhat technical, finding of this work is that the chosen roughness metric decreases linearly with model simulation time. Previous studies have found an exponential age v. roughness relationship, which means that absolute uncertainties of predicted ages are quite large for the older landslides on the steeper part of that curve. A relatively simple fix of using this or a similar roughness metric may dramatically reduce the uncertainty on predicted ages of old landslides.*

AC: This is an excellent point that we hadn't thought about. Since the model of MADstd values only applied diffusion to a landscape, we would be hesitant to predict a linear relationship over long time scales. As addressed in our Referee #2 response, we plan to do more diffusion models

of MADstd to see how the roughness values vary with diffusion rate, and what a reasonable range of MADstd values with age is. We could include some runs with stream erosion and see if the MADstd-age relationship continues to be linear. In any case, following Referee #2's suggestions, we plan to expand the discussion section to include limitations of the MADstd technique. We plan to incorporate Referee #1's note that incorporating this technique or a similar metric to landslide studies could increase the accuracy of predicted ages for older landslides.

**Response to Referee #2:**

*RC2: One thing I found confusing is the discrepancy between the model MADstd values and the MADstd values for earthflows in the Teanaway basin. I don't understand how mid-Holocene ages were estimated for the older earthflows in the Teanaway when MADstd values are much lower than the lowest in the model. Perhaps this can be expanded. Unlike reviewer #1, I don't think it matters that there is poor correlation between the absolute ages presented here and the MADstd – I think it's clear how much uncertainty there is in the earthflow ages, particularly the older ones. That said, I think it might be nice to show in a figure that where you do have field evidence of relative ages (e.g., cross-cutting relationships), it does work. Also, I think this part of the discussion should be moved to the results.*

AC: We plan to expand the discussion of the MADstd values and estimated ages. We assumed that MADstd values would vary based on diffusion rates and advective processes acting alongside the earthflow; we assumed this would mean the model MADstd and actual MADstd values may not match, thus the apparent logic leap when we estimated a mid-Holocene age. However, we plan to test this assumption; for one, we can run the MADstd model again with different diffusion values to estimate a likely range in MADstd values with earthflow age. This would allow us to better support an age estimate for the Teanaway earthflows, whether that age ends up being mid-Holocene or not. We also plan to shift the focus of the manuscript away from concrete dates/timing of earthflows into a focus on earthflow controls and morphologic effects.

We plan to add a figure that shows the cross-cutting relationships, and move this discussion section to the results.

*RC2: There is too much focus on salmon habitat. Although I think it's a good motivation for investigating the timing and controls on earthflows and how they impact valley width, the focus on it (e.g., section 2.2) implies there will be a solid conclusion relating to it. In the end, the claim is that given that earthflows were active when salmon populations stabilized, they don't seem to have negatively impacted salmon habitat. It's not a big conclusion, but reviewer 1 may be right that the data still doesn't support it. We can never know what fish populations would be if there were no earthflows. I think all you can say is that earthflows contribute to topographic heterogeneity and that non-catastrophic disturbances and topographic heterogeneity are generally good for biodiversity.*

AC: Thank you for this point. The study was originally conceived as a senior thesis related to salmon habitat, and we got stuck on this point without stepping back and assessing whether it was accurate. We plan to revise the manuscript to focus on the structural controls and geomorphic implications of the earthflows, rather than focusing on salmon habitat. We do think habitat disturbance is an interesting implication and perhaps motivation for further work, but we plan to confine this to a smaller section in the discussion that does not offer definitive conclusions about habitat and earthflow timing, but rather speaks generally to the topographic alterations being good for habitat.

*RC2: The hypotheses about climate control on earthflow activity seem like a bit of a stretch given the uncertainty on earthflow ages. But, if you're going to discuss this, Bennett 2016 probably ought to be referenced (see below). Perhaps though, instead, more focus should be on the main results: earthflows are active in the Teanaway basin, are structurally controlled, and act to modify valley width and hence floodplain habitat, as well as sediment flux and most likely grain size. Much of the discussion could instead be used to discuss limitations on the techniques employed and areas of future work.*

AC: We plan to shift the focus to be on the active earthflows in the Teanaway, the structural control, and the valley width impacts. Our discussion will be revised to focus on impacts of the earthflows on sediment flux, grain size and habitat, as well as a discussion of the limitations of the techniques and areas of future work. This will shift focus away from the climate control, which we agree is a tenuous connection.

*RC2: I suggest deleting everything about rotational and translational slides to better focus on earthflows.*

AC: We plan to remove the sections about rotational and translational slides to focus the study on earthflows.

*RC2: Figure comments*
*Fig. 3 I think this could be moved to a supplement*
*Fig. 7 I think this could be moved to a supplement*
*Fig. 8 Why isn't valley width plotted against discharge as in May, 2013? It would also be really nice to see some lidar hillshade images of these constrained and upstream widened reaches.*
*Fig. 9 I think this figure should be moved up to the methods or results*

AC: We plan to add a figure showing the cross-cutting relationships that support the MADstd values. We will move figures 3 and 7 to a supplement, or remove them altogether. We plan to move Figure 9 as well as the discussion of MADstd ages to the results section. For Fig 8 and the valley width results, we will add images showing the constrained and widened reaches. We will plot valley width against drainage area (as a proxy for discharge) to better match previous work.

---

## Author Response (AR1)

Dear Editor,

Thank you and the two anonymous reviewers for constructive and critical comments on our submitted manuscript. In the revised version, we addressed your comments through a major revision and have altered the manuscript focus--and title--to be on lithologic controls on earthflows, the effect of earthflows on valley width and sediment, and MADstd as a potential method for relative dating earthflow activity. We believe these revisions addressed your and reviewers' concerns, particularly regarding the relationship between MADstd and absolute age and regarding the historic analysis of salmon habitat. We are confident these revisions will improve the quality of the manuscript.

Our detailed comments are below. Please note that line numbers refer to the track-changes document.

Sincerely,
Sarah Schanz, on behalf of the authors

**Reply to Referee #1**

*RC1: First, to test the hypothesis that high earthflow frequency corresponds to times of large salmon populations, the salmon population would need to be independently known over the Holocene. I'm not a salmon expert, but that information doesn't seem readily attainable, and the manuscript doesn't provide any independent references that have determined it. Instead, there is a suggestion that salmon populations may have stabilized 4-5 ka based on very broad scale inferences about climatic and tectonic changes.*

AC: We agree that this hypothesis cannot be readily tested by the data. We restructured the goals of the papers to focus on controls on earthflow activity. Salmon habitat is mentioned as a motivation for understanding the impacts of earthflows, and as a potential implication of the Teanaway earthlows' effect on sediment loads and valley widths. These changes can be seen most prominently in the introduction, where previous paragraph #2 has been deleted; background where the section on historic salmon habitat disturbance is deleted; and the discussion section 5.2 which now contains all of our implications for salmon habitat and is much shorter than the original manuscript.

*RC1: Second, the study did not present a significant, data-derived relationship between earthflow age and surface roughness, which would be necessary in order to use surface roughness to infer age of undated earthflows.*

*Age vs. activity. Since age (~time since a landslide happened) and activity (~how active the landslide is currently) both affect a landslide's surface roughness, it can be quite challenging to disentangle these two effects for earthflows in particular. This may be one reason why there is not a relationship between landslide age and roughness for the study site. Fig. 5 is a bit*

*misleading on this point, since it shows a modeling result, which by design will smooth the landslide deposit over time. If you add the age and roughness values from Table 1 to the figure, it is clear that the radiocarbon-based ages and roughnesses are not correlated and do not follow the model-predicted trend. Those data are so scattered that that general inferences about mid or late Holocene timing aren't supported either (e.g. the smoothest landslide is the youngest (Dickey Cr.) and the oldest is the roughest (Indian Cr.)). Perhaps the fact that age and roughness don't correlate is a useful observation for thinking about what other factors control surface roughness of these earthflows.*

*Landslide dates. As stated in the manuscript, charcoal provides only a maximum age, and often this age is not a close maximum. E.g. see Struble et al. (2020), GSA Bull. where charcoal from a landslide's deposit is up to thousands of years older than the landslide. Also in the present manuscript, the discrepancy between the minimum and maximum ages of the Rye Creek earthflow (346 and 4,353 yrs, respectively) suggests that either the earthflow's age isn't accurately enough known to determine an age roughness model, or that a single age for the earthflow may not be representative of its long term behavior, such as persistent movement and/or reactivations.*

AC: The distinction between age and activity is a good point, and below we show how we edited the manuscript to make this clear. We also show how we edited the manuscript in recognition that the MADstd method is limited by our absolute age controls.

We added the direct comparison to Struble et al. (2020) in lines 302 and 564 where we acknowledge the potential error in radiocarbon ages [line 302] and compare our sedimentation ages and radiocarbon ages to show ~4000 years over-estimation in the radiocarbon ages. This is similar to values found by Struble et al. (2020) [line 564].

We revised to make it clear what each dating method implies. For radiocarbon and sedimentation ages, we altered the heading to show that these are maximum ages [line 294 "3.1 Earthflow mapping and maximum earthflow ages; line 521 "4.2 Maximum earthflow ages"]. We added clarification of what these ages represent in the methods section [lines 301-305 for radiocarbon; lines 319-320 for sedimentation] and again in the results [lines 555-567].

For MADstd, we revised the manuscript to make it clear that time since last major activity is being evaluated; and that the MADstd metric gives relative earthflow activity, not relative age. This revision can be found in the headings [lines 365 and 815] and in the methods, results, and discussion sections discussing MADstd [lines 366-374 for methods; 577-835 for results; and 905 for discussion].

We also shifted the focus of the manuscript away from salmon habitat and timing of earthflow activity. We recognize we did not have enough information to attempt this before. Instead, following Referee #2's suggestions, the manuscript was edited to focus on lithologic controls on earthflows, impacts of earthflows on valley width and sediment, and introduction and discussion of a potential method to rate earthflow activity (but we make it clear that we are not

tying MADstd directly to absolute dates/ages). We believe these changes addressed Referee #1's concerns about the validity of using MADstd to infer ages and inferences about earthflow timing.

The changes above can be seen throughout the manuscript but especially in:
- Title has been altered to focus on controls on earthflow activity rather than salmon habitat and relative dating.
- Background section focuses on the study site geology and the portions regarding salmon habitat and disturbances have been deleted.
- Results [lines 425 – 491] emphasize the position of earthflows and lithologic relationships.
- Discussion is now organized into three sections on: Drivers of earthflow motion [lines 836-876] and Landscape disturbance [lines 877-903]. Briefly in section 5.2 Landscape Disturbance, we hypothesize some habitat disturbance in the form of valley width changes and sediment loads, but we do not focus on salmon and avoid making absolute conclusions about the timing of salmon habitat change.
- We also altered discussion of the timing of earthflows to be much looser. Recognizing that our absolute ages are mostly in the last few hundred years, we do not extrapolate MADstd beyond those bounds. We briefly hypothesize that anthropogenic activity may have contributed to recent earthflow activity, based on timing and a lack of a strong climate signal [lines 868-876].

*RC1: An exciting, if somewhat technical, finding of this work is that the chosen roughness metric decreases linearly with model simulation time. Previous studies have found an exponential age v. roughness relationship, which means that absolute uncertainties of predicted ages are quite large for the older landslides on the steeper part of that curve. A relatively simple fix of using this or a similar roughness metric may dramatically reduce the uncertainty on predicted ages of old landslides.*

AC: This is an excellent point that we hadn't thought about. We added stream erosion to our diffusion model – this is updated in the code stored on Github – and we ran a suite of models with varying diffusion rates and with and without stream power [lines 383-391]. We found that stream erosion did not change MADstd [lines 577-580], and that all diffusion runs result in a linear relationship with roughness and simulation time. The old Figure 5, which is now Figure 6, is updated to show a range of diffusion values.

We added a paragraph to our discussion of MADstd that discusses the potential for this linear model to add precision to surface roughness age methods [lines 970-977].

**Response to Referee #2:**

*RC2: One thing I found confusing is the discrepancy between the model MADstd values and the MADstd values for earthflows in the Teanaway basin. I don't understand how mid-*

*Holocene ages were estimated for the older earthflows in the Teanaway when MADstd values are much lower than the lowest in the model. Perhaps this can be expanded. Unlike reviewer #1, I don't think it matters that there is poor correlation between the absolute ages presented here and the MADstd – I think it's clear how much uncertainty there is in the earthflow ages, particularly the older ones. That said, I think it might be nice to show in a figure that where you do have field evidence of relative ages (e.g., cross-cutting relationships), it does work. Also, I think this part of the discussion should be moved to the results.*

AC:
We expanded the discussion of MADstd values and estimated ages by adding a section in the results called "4.4 Verification of MADstd relative dating" [lines 576-811]. This section contains much of the content previously in the manuscript discussion.

We re-ran the model with different diffusion values and with advective processes (stream power) (see response to RC1 comments above). However, this showed that for low diffusion values, the low slope between MADstd and age results in a high range of ages for a single MADstd value, if you do not know the diffusion rate precisely (Figure 6). Based on this, we revised the manuscript to avoid making conclusions about earthflow ages in the mid Holocene. Instead, we focus on MADstd values we have absolute age controls for, which are in the last few hundred years [lines 815-835].

In the Verification of MADstd… section, we discuss the range of MADstd values for similar earthflow activity age [lines 711-811]. Through this revision, we hope we address RC2 and RC1's concerns that MADstd is too heavily extrapolated.

We created a new figure to show cross cutting relationships [Figure 7] as well as a summary of our absolute ages and MADstd ages [Figure 5].

*RC2: There is too much focus on salmon habitat. Although I think it's a good motivation for investigating the timing and controls on earthflows and how they impact valley width, the focus on it (e.g., section 2.2) implies there will be a solid conclusion relating to it. In the end, the claim is that given that earthflows were active when salmon populations stabilized, they don't seem to have negatively impacted salmon habitat. It's not a big conclusion, but reviewer 1 may be right that the data still doesn't support it. We can never know what fish populations would be if there were no earthflows. I think all you can say is that earthflows contribute to topographic heterogeneity and that non-catastrophic disturbances and topographic heterogeneity are generally good for biodiversity.*

AC: Thank you for this point. The study was originally conceived as a senior thesis related to salmon habitat, and we got stuck on this point without stepping back and assessing whether it was accurate! We revised the manuscript to focus on the structural controls and geomorphic implications of the earthflows, rather than focusing on salmon habitat. We removed salmon habitat from the background section and removed the discussion section on it. We retained

habitat as a motivation for understanding earthflow controls, seen in the introduction, and we briefly address how the changes in valley width and sediment loads might affect habitat [lines 896-903]. However, the broader discussion section [lines 877-895] focuses on larger-scale topographic disturbance rather than narrowly on salmon habitat history.

*RC2: The hypotheses about climate control on earthflow activity seem like a bit of a stretch given the uncertainty on earthflow ages. But, if you're going to discuss this, Bennett 2016 probably ought to be referenced (see below). Perhaps though, instead, more focus should be on the main results: earthflows are active in the Teanaway basin, are structurally controlled, and act to modify valley width and hence floodplain habitat, as well as sediment flux and most likely grain size. Much of the discussion could instead be used to discuss limitations on the techniques employed and areas of future work.*

AC: We shifted the focus to be on the active earthflows in the Teanaway, the structural control, and the valley width impacts. Our discussion was revised to focus on impacts of the earthflows on sediment flux, grain size and habitat, as well as a discussion of the limitations of the MADstd techniques and areas of future work.

The changes above can be seen throughout the manuscript but especially in:
- Title has been altered to focus on controls on earthflow activity rather than salmon habitat and relative dating.
- Background section focuses on the study site and the underlying geology and the portions regarding salmon habitat and disturbances have been deleted.
- Results [lines 425 – 491] emphasize the position of earthflows and lithologic relationships.
- Discussion is now organized into three sections on: Driver of earthflow motion [lines 836-876] and Landscape disturbance [lines 877-903]. Briefly in section 5.2 Landscape Disturbance, we hypothesize some habitat disturbance in the form of valley width changes and sediment loads, but we do not focus on salmon and avoid making absolute conclusions about the timing of salmon habitat change.
- We also altered discussion of the timing of earthflows to be much looser. Recognizing that our absolute ages are mostly in the last few hundred years, we do not extrapolate MADstd beyond those bounds. We briefly hypothesize that anthropogenic activity may have contributed to recent earthflow activity, based on timing and a lack of a strong climate signal [lines 868-876].

*RC2: I suggest deleting everything about rotational and translational slides to better focus on earthflows.*

AC: We removed the sections about rotational and translational slides to focus the study on earthflows. This is seen in an edited Figure 2 and Figure 3 which now only show earthflows.

*RC2: Figure comments*

*Fig. 3 I think this could be moved to a supplement.*
We removed Figure 3, as the results could be summarized in one sentence. [lines 486-489]

*Fig. 7 I think this could be moved to a supplement*
We removed Figure 7. In our revised version, we did not find it necessary to discuss earthflow area vs MADstd.

*Fig. 8 Why isn't valley width plotted against discharge as in May, 2013? It would also be really nice to see some lidar hillshade images of these constrained and upstream widened reaches.*
We edited the figure, now Figure 4, to show valley width against upstream drainage area, as a proxy for discharge. We also added three hillshade images to show typical narrowed reaches.

*Fig. 9 I think this figure should be moved up to the methods or results*
We deleted Figure 9 because we added two figures showing the cross-cutting relationships [Figure 7] and summarizing the absolute age controls [Figure 5]. Figure 5 has an inset map showing the MADstd of an earthflow complex and serves the same purpose as Fig 9 in showing the uniform MADstd across a forested hillslope, and the higher MADstd values near scarps.

---

## Author Response (AR2)

Dear editor,

Thank you and the two anonymous reviewers for the second round of feedback on the manuscript. The comments were helpful in clarifying the manuscript, particularly in the methods, and in refining the focus to earthflow influence on the landscape rather than on habitat. Below, we have detailed our response to the referee comments.

Sincerely,
Sarah Schanz, on behalf of the authors

~~

*As mentioned I found the introduction quite confusing as it focuses on the impact of earthflows on salmon habitat rather than earthflows themselves. As the primary outcome of the paper is a tool which describes the activity of the earthflows and identifies their distinctive features I feel the introduction needs more reference to the earthflow literature.*

The reference to salmon habitat unfortunately lingered after the first revision. We revised the first paragraph of the introduction, where habitat was mostly mentioned, to focus more on geomorphic impacts. We tried to maintain a balance in which habitat was still mentioned as a motivating factor for research into earthflows – even if we can't concretely address it here. To that end, we made habitat a vaguer implication. We added more background information on earthflow response to climate.
Clean doc: Lines 23-324, 43-45.
Track changes doc: 23-44, 55-57.

*The study area section is very focused on the geology of the region rather than any known impacts of the earthflows (or other landslides) on the landscape. I would also expect a bit more discussion on the human impact of the area if known i.e. how much of the region was logged? What is the coverage these days? I would also expect some discussion of whether salmon spawn in the region if most of the study is focused on the impact of earthflows on salmon habitat.*

In the first paragraph of the study area, the logging history is discussed. We expanded this to state the impact of logging on rivers, and that all forests are <100 years old in the study area. Widespread logging practices ceased in the 1940s, though patchwork logging does occur. We retained the geology focus because the geology controls where earthflows occur, and so is important to understand. Since we have revised the introductions to de-emphasize salmon habitat, at the suggestion of the first round of reviews and the above comment, we don't include a discussion of whether salmon spawn in the region.

We don't know of any studies detailing the impacts of earthflows and other landslides on this particular landscape, and have made that clear in the last paragraph of the study area section. The North Fork Teanaway Watershed Analysis did examine shallow landslides near stream banks, but only with the aim of roughly estimating potential habitat threats to fish. No impacts were quantified or mapped.
Clean doc: Lines 79-81, 106, 110-111.
Track changes doc: Lines 105-107, 132, 137.

*The current structure of the results section is a bit confusing as it jumps between subjects. I would suggest that the mapping results should be followed by a discussion of the MADstd trends, then the Maximum ages (radio carbon dating) and then the relative dating and verification of the dates. This will help the reader to follow the logic of the MADstd metric and its use in relative dating a little easier.*

We appreciate the referee's suggestion. However, we prefer to keep the earthflow verification section before relative activity results, so that the relative activity results are based on a MADstd system that is shown to work. We do recognize that the apparent subject jumps are confusing, particularly without any overarching logic laid out. To that end, we have added an introduction/roadmap paragraph to the results that explains why the results are ordered the way they are.
Clean doc: Lines 220-223
Track changes doc: Lines 322-325

*In the methods section, I think the expected power law scaling should be presented. See two recent Esurf preprints about valley width for good examples - one by Clubb et al. and one by Harel et al. In the results section, I think you should fit power laws to the data and show the best-fit equations on figure 4 (even if they are bad fits, which is expected given the influence of earthflows).*

We added text about the expected power law scaling, including an example from similar lithologies in the Pacific Northwest, USA, and the suggested example from Clubb et al in the Appalachian Mountains, USA. Both examples have a power law exponent of 0.3, and we added this regression line to the background of Figure 4 for reference. Interestingly, the non-landslide points do match this regression exponent, though the earthflows clearly do not.
Clean doc: Lines 198-204, Figure 4 and caption
Track changes doc: Lines 270-275, Figure 4 and caption

*Figure 7 is a bit confusing at first. I didn't notice that the outlines of B, C, and D were colored at first and I wasn't sure what the colors of the earthflows meant. I think you need a MADstd color bar and I think you should explain in the caption what B, C, and D are examples of. The caption is short now so I think you have room to spell it out clearly. Also, D looks like an example of a case where MADstd and topography match?*

Taking into account both referees' comments to Figure 7, we revised panels B-D to be clearer. First, we added a line to clearly connect each panel to the associated bar in Panel A. We added MADstd labels to each earthflow and clarified the color scheme in the caption. For panel D, we zoomed in to show this better. None of the examples of MADstd and topography NOT matching are very clear, and this is the best example we could show. We expanded the figure caption to explain B-D, and to point the reader to the toe of the landslide in D that is onlapping a higher MADstd earthflow.
Tracked changes and clean doc: Figure 7 and caption.

| Reviewer comment | Line change (clean) | Line change (tracked) | Author comment |
|---|---|---|---|
| Line 46: It is currently not clear where the Eel River Basin is or how any study there may relate to the study area. | 50 | 62 | We clarified that the Eel River is in California, USA, and that it is used here as an example of how earthflows can influence landscape evolution. |
| Line 48-49: This sentence is confusing and hard to follow | 52-54 | 64-66 | Revised to: "Additionally, in-stream sediment production from earthflows is unsteady because annual to decadal precipitation conditions cause intermittent movement over the decades to |

| | | | centuries that the earthflow is active." |
|---|---|---|---|
| Line 51: Again, the Eel River is referenced as if the reader should know where it is or why it is an important reference for this study. It is also not clear how earthflows produce topographic highs from this sentence. | 55-58 | 67-69 | Added 'California, USA' after Eel River to give location, and restructured the sentence to both clarify that this is an example of where earthflows influence topography and clarify how topographic highs are produced from differential erosion:

"Differential erosion by earthflows results in valley-scale topographic patterns: lithologic controls on earthflow location in the Eel River basin, California, USA, concentrates erosion in the melange units, and lowers that landscape relative to isolated resistant sandstone outcrops." |
| Figure 1: The regional bedding information should be included on the map as it is relevant to the results of the study. | Fig 1 | Figure 1 | We've added the regional bedding in the form of strike and dip symbols; we did not include dip angle because it made the map unreadable. |
| Line 114: How were the landslides mapped? Was this done by hand? How were the edges of the earthflows delineated? Typically, landslides are identified from satellite images by changes of colour but it is not clear how this is done with the Lidar data. | 123, 126-127 | 151, 154-155 | We clarified that earthflows were mapped in ArcGIS and edges were identified by the morphologic shear zones, depositional toes, or erosional scarps.

"We mapped the edges of earthflows as the edge of shear zones next to undisturbed hillslopes, and used scarps and toe deposits to delineate the top and bottom of earthflows from surrounding hillslopes." |
| Line 115: What is meant by hourglass shape? Does this mean the scar and toe of the landslide are wider than the centre? | 124 | 152 | Yes, it does mean that. We clarified for others unfamiliar with the term (which we realize now is probably used mostly in US-centric textbooks and is not globally common):

"..hourglass shape with a wide head scarp and toe compared to a narrow transport zone.." |

| Line 116-117: Are these older earthflows undistinguishable from the rest of the landscape or from other mass movements? | 127-130 | 156-158 | It is both, but the latter point is important as we want to focus only on earthflows. Added/edited text:

"These morphologic clues degrade over time and it becomes harder to distinguish earthflows from other mass movements. Therefore, we focus our analysis on Holocene earthflow activity when it is still possible to distinguish the characteristic earthflow morphologies." |
| --- | --- | --- | --- |
| Line 120: How were the dated earthflows selected? | 136-138 | 164-166 | This is detailed in the middle of the paragraph. The existing line was edited to clarify what a visual estimate of roughness is:

"Sampled earthflows were selected based on potential for a fresh exposure via road or stream erosion and to capture a range of activity, which we estimated by the prominence of levees and shear zones in the bare earth lidar" |
| Line 123: Maximum earthflow activity does this refer to the duration of the earthflow activity? Activity and age seem to be used interchangeably throughout the manuscript but they may mean different things to other readers. | 135-136 | 163-164 | We added a sentence to clarify that this is when earthflow activity started.

"Maximum earthflow activity refers to a maximum estimate of how long since an individual earthflow first became active." |
| Line 124: What is a visual estimate of roughness? | 136-138 | 164-166 | Clarified to be specific:

"Sampled earthflows were selected based on potential for a fresh exposure via road or stream erosion and to capture a range of activity, which we estimated by the prominence of levees and shear zones in the bare earth lidar" |
| Line 133: As this methodology and the ages of the lakes are an important constraint on the aging of the | 145-150 | 181-184 | In addition to the reference to the methods outlined in Struble et al (2020), we summarized their methods to extract volume |

| | | | |
|---|---|---|---|
| earthflows this methodology needs to be explained. | | | in two new sentences and an edited existing sentence: "We estimated the valley bottom elevation under the lakes using the average valley slope of surrounding un-dammed valleys. This valley floor estimate is interpolated in GIS with the lake perimeter elevation to form an estimated lake bottom topography. The bottom topography is subtracted from the lidar surface elevation to estimate the sedimentation volume post-earthflow." |
| Line 135: What does similar mean here? Are these basins in the same region as the study area? How were these rates produced? | 150-151 | 184-185 | We specified that the basins are neighboring and added that the rates are produced by beryllium 10 erosion dating. We suggest Moon et al. (2011) for further details on how the rates were produced, but re-arranged this reference to make it clear that denudation rates come from Moon et al. "We used nearby mid-Holocene $^{10}$Be denudation rates of 0.08 and 0.17 mm/yr (Moon et al., 2011) from neighbouring basins with similar mean annual precipitation and glaciation" |
| Lines 135 – 138: Include the equation and calculations to produce these dates. Currently it is not very clear what is happening. | 151-154 | 185-188 | We added/revised the prior section to step through our calculations. "We multiplied the denudation rates by upstream contributing area for each lake to give a volumetric estimate of sediment delivery per year. The lake sedimentation volume is divided by this rate to estimate the years necessary to fill each lake. We repeat this process with the upper and lower denudation bounds to give a range of plausible sedimentation ages, which |

| | | | approximate when the earthflow dammed the creek." |
|---|---|---|---|
| Line 142: A figure (maybe in the supplemental) of what an earthflow looks like in Lidar would be helpful. Could also be part of a multi-panel showing the different steps used to produce the MADstd of the earthflow. | Supplement Figures 2, 3 | Supplement Figures 2, 3 | Created two diagrams showing how MADstd was produced. Incudes figures of an earthflow as it goes through the process. This figure is included as Supplemental Figures 2 and 3. |
| Lines 144 – 146: It is not clear what MAD is a measure of. The referenced paper and the given descrtiption are too specialist for your typical reader. | 162-164 | 196-198 | We added more to the description, both in the section pasted below but also in our description of flow direction that can help the reader understand MAD. The referenced paper (Trevisani and Cavalli, 2016) does give good visual examples of MAD in hillslope and valley terrains, though their equations are very technical.

"MAD is a bivariate geostatistical index that analyzes residual elevations between paired locations in a Digital Elevation Model (DEMs) (Trevisani and Cavalli, 2016). MAD results in a roughness index for each direction (N-S, E-W, NE-SW, and NW-SE) across each raster cell in the study area." |
| Lines 146-147: How is MAD combined with flow direction? | 164-166, Supplement figures 2-3 | 198-200, Supplement figures 2-3 | We have added a sentence describing this:

"Using surface flow directions derived from the DEM surface, these directional roughnesses are filtered to correspond to the flow direction; e.g., if the surface flow direction is N-S, then only the N-S roughness is used (Trevisani and Cavalli, 2016)." |
| Line 150: In line 140 MAD is said to be a measure of earthflow activity and now we are looking at the relationship | 179-181 | 259-262 | Thank you for catching that error. Age should be activity, and we've edited to say activity. |

| | | | |
|---|---|---|---|
| between age and MAD. Are activity and age being used interchangeably? | | | |
| Lines 150 – 165: It is not clear why these models are being applied to the DEM. | 118-119, 179-181 | 146-147, 259-262 | The models are being applied so that we can test the assumption that MADstd decreases with time since last earthflow activity. Diffusion in the model helps us simulate this.

We added lines to clarify.

*"The MADstd relative dating method rests on the assumption that earthflow MADstd will decrease with time since last earthflow activity due to soil diffusion. In order to test this assumption, we simulated landscape diffusion on a recent earthflow and calculated MADstd through time; if our assumption is correct, MADstd should decrease with simulation time."* |
| Line 153: Why does it matter that the stream is not permanently dammed? | 184 | 264 | It does not matter, and we have removed this phrase. |
| Lines 167 – 174: This paragraph should be higher up in the section before the discussion of diffusion models. We also need a clear definition of what directional roughness is. | 169-177 | 203-257 | Paragraph has been moved up, and directional roughness has been explained in response to an earlier comment by the referee. |
| Line 180: How is width measured? It is not clear from the current description. | 211-212 | 313-314 | Width is measured as the length of the transect within the defined valley bottom. We added a phrase to clarify:

"..measure valley width *as the width of the transect line within the 5% valley slope.*" |
| Line 189: Were only earthflows mapped? If so this section could be called earthflow mapping to avoid confusion. Reporting the basic statistics of earthflows would be useful, i.e. smallest and largest | 224, 225-226 | 326, 327-328 | We changed the section name to 4.1 *Earthflow* mapping.

We added a sentence with earthflow statistics:
*"Earthflows range in size from 1076 $m^2$ to earthflow complexes* |

| earthflows (area or length), average size etc. | | | 4e6 $m^2$ in area with a median area of 28,547 $m^2$." |
|---|---|---|---|
| Line 219: If possible an extra panel showing how the width of tributaries unaffected by earthflows changes with drainage area would be helpful. | Figure 4 | Figure 4 | We added a regression line in each panel that shows the drainage area-valley width relationship for valleys unaffected by landsliding (Schanz and Montgomery, 2016; Clubb et al., 2022) |
| Figure 5: Unclear what Inset is meant to be showing. There needs to be a color bar and a label or legend highlighting the lake. | Figure 5 | Figure 5 | We have added a label to the lake as well as a color bar for the 5m MADstd. |
| Line 247: The earthflow could have taken some time to dam the channel | 285 | 391 | This is a good point. We have added clarification that this may not be exactly a maximum age estimate.

"..represents a maximum age – or a near maximum age if earthflow velocity was slower and did not immediately dam the channel." |
| Line 252 – 257: How can the residence time of carbon in earthflows be 4000 years when no other landslide comes close to that in terms of age? | -- | -- | We state that the residence time can be upwards of 4000 years; we don't expect all radiocarbon samples to have this large of residence time. And as explained in the text (line 274 clean), residence time can come from inherited time in the charcoal, not from earthflow movement/storage. And we do have two of our six samples that are greater than 4000 radiocarbon years. |
| Lines 268-272: This hypothesis was not stated earlier in the manuscript. | 179-181, 306 | 259-262, 414 | We clarified. This is the assumption that underlies the roughness-age correlation. This is made clearer in the methods and is also restated here:

"Simulated diffusion verified our assumption that MADstd decreases with time since earthflow activity." |

| | | | |
|---|---|---|---|
| Lines 265 – 272: It still isn't clear why this experiment and model was done. | 118-119, 179-181, 306 | 146-147, 259-262, 414 | We have edited the methods to clarify this, and added a line to remind the reader of why the simulation was done (see comment just above this). |
| Line 275: The properties of the earthflow could also affect the metric, slower flows may have lower MADstd values | 316-317 | 425-426 | This is a good point, and would affect the relative degree of diffusive and advective processes. We added a phrase to show that this is another way in which we do not understand the diffusive history:

"..the soil diffusion value, and on earthflow velocity, which would affect the relative strength of diffusive versus advective processes." |
| Line 280: Is it possible to assume a K value from the dated earthflows? If an initial roughness is assumed a trend can be found and perhaps the modelling can provide an estimate for the K value. | -- | -- | Through inverse modeling, we could try to estimate K from the earthflows where we have a good age control. However, as the referee pointed out earlier, the morphology of the earthflow can also be affected by the velocity of the earthflow so initial roughness wouldn't be the same across all earthflows. |
| Line 285: This isn't quite clear, do the flows being cut have lower MADstd values? | 326 | 435 | We added a clarification that the cut flows have lower MADstd values.

"Of these, 16 had MADstd values consistent with the cross-cutting relationship where the on-lapping earthflow had higher MADstd values than the underlying earthflow." |
| Figure 7 B-D. it is unclear what is happening in these panels. The polygons could be labelled with their MADstd values and the cross-cutting relationships can be made more obvious. | Figure 7 | Figure 7 | Polygons have been labelled with MADstd values to make the relationships more obvious, and lines now link each panel to the category in panel A that they represent. Figure caption is expanded. |
| Lines 294-299: This is only true if there is clear evidence of erosion of the toes of the older flows | 343-344 | 456-57 | That is true, and we see evidence for erosion back to the valley walls in Figure 4, where low MADstd earthflows (light pink colors) fall along the |

| | | | regression line of non-landslide terrain. We added a sentence: "Earthflows with low MADstd values are near the regression line for non-landslide terrain, suggesting their toes have been eroded to the valley walls (Figure 4)." |
|---|---|---|---|
| Figure 8: Could have activity threshold marked on the histograms | Figure 8 | Figure 8 | Added a grey bar to the background of the histograms that shows the 0.13-0.15 MADstd activity level. |
| Line 350: Is there a preferential rainfall or wind direction? | -- | -- | We could not find this data. In the field, we don't see a strong aspect dependence for vegetation, which we would expect to see if there is a preferential rainfall direction or even wind direction. |
| Line 398 – 400: Is there a photo of the field observations? | Supplement Figure 1 | Supplement Figure 1 | Images of the toes of earthflows are now included in the supplemental information as Supplemental Figure 1. |
| 322 typo - primarily should be primary | 372 | 487 | Corrected to primary. |
| 365 the extent of earthflow activity? | 417 | 534 | Replaced 'glaciation' with 'earthflow activity' |
| 373 Probably add Bennett 2016 to this sentence somewhere - https://doi.org/10.1002/2016GL068378 | 43-45, 116, 425 | 55-57, 144, 542 | Added Bennett 2016 reference w/regards to climate correlation, and added to the introduction. |
| 404 change 'slopes' to 'channel slopes' | 457 | 575 | Inserted 'channel' before 'slopes' |